# GPCR Binding and JNK3 Activation by Arrestin-3 Have Different Structural Requirements

**DOI:** 10.3390/cells12121563

**Published:** 2023-06-06

**Authors:** Chen Zheng, Liana D. Weinstein, Kevin K. Nguyen, Abhijeet Grewal, Eugenia V. Gurevich, Vsevolod V. Gurevich

**Affiliations:** Department of Pharmacology, Vanderbilt University, Nashville, TN 37232, USA; chen.zheng@vanderbilt.edu (C.Z.); liana.d.weinstein@vanderbilt.edu (L.D.W.); knguye35@uthsc.edu (K.K.N.); abhijeet.grewal@vanderbilt.edu (A.G.); eugenia.gurevich@vanderbilt.edu (E.V.G.)

**Keywords:** arrestin, GPCR, JNK3, conformation, signaling bias

## Abstract

Arrestins bind active phosphorylated G protein-coupled receptors (GPCRs). Among the four mammalian subtypes, only arrestin-3 facilitates the activation of JNK3 in cells. In available structures, Lys-295 in the lariat loop of arrestin-3 and its homologue Lys-294 in arrestin-2 directly interact with the activator-attached phosphates. We compared the roles of arrestin-3 conformational equilibrium and Lys-295 in GPCR binding and JNK3 activation. Several mutants with enhanced ability to bind GPCRs showed much lower activity towards JNK3, whereas a mutant that does not bind GPCRs was more active. The subcellular distribution of mutants did not correlate with GPCR recruitment or JNK3 activation. Charge neutralization and reversal mutations of Lys-295 differentially affected receptor binding on different backgrounds but had virtually no effect on JNK3 activation. Thus, GPCR binding and arrestin-3-assisted JNK3 activation have distinct structural requirements, suggesting that facilitation of JNK3 activation is the function of arrestin-3 that is not bound to a GPCR.

## 1. Introduction

Arrestins were discovered as key players in homologous desensitization of G protein-coupled receptors (GPCRs) [1]. Most vertebrates, including humans, express four arrestin subtypes [2]. Two of these, arrestin-2 and -3 (a.k.a. β-arrestin 1 and 2, respectively), are ubiquitously expressed. Both bind hundreds of different GPCRs, with arrestin-3 being the most promiscuous of the two [3,4]. Arrestins not only suppress G protein activation by GPCR but also serve as signal transducers, facilitating several branches of signaling [5,6]. The best-known signaling function of arrestins is activation of the mitogen-activated protein kinase (MAPK) pathways [7,8]. Both non-visual subtypes, arrestin-2 and arrestin-3, facilitate the activation of ERK1/2 [8], whereas only arrestin-3 enhances the activation of the JNK family kinases [7,9]. The classical paradigm of arrestin-mediated signaling posits that the receptor-bound arrestin shuts down G protein activation and at the same time initiates the second round of G protein-independent signaling [5]. This paradigm implies that the binding of arrestin to a GPCR is required for arrestin-dependent signaling to occur, with arrestin redirecting the GPCR signaling from G protein-dependent to arrestin-dependent pathways [10]. This is true for the ERK pathway: the affinity of free arrestin for ERK1/2 is very low, and arrestin-assisted activation of ERK requires a GPCR input [8,11,12]. A significant body of evidence suggests that GPCR activation affects arrestin-dependent signaling via Src [13,14] and focal adhesion kinase [15]. In contrast, several lines of evidence indicate that arrestin-3 activates MAPKs of the JNK family in a GPCR-independent manner. First, free arrestin-3 binds JNK3, its preferred JNK isoform, quite well [16,17]. Second, in contrast to ERK1/2 activation in the same cells, GPCR stimulation has no effect on the arrestin-3-dependent activation of JNK3 [12]. Third, arrestin-3-derived peptides incapable of binding GPCRs facilitate activation of JNK3 in cells [18,19]. Arrestins in their basal conformation independently of GPCRs bind many proteins, although in most cases the signaling consequences of these interactions have not been elucidated [20,21,22,23,24].

Biased GPCR signaling via one of the two classes of transducers, G proteins or arrestins, has recently attracted considerable attention due to its potential to retain desired therapeutic efficacy while minimizing unwanted on-target side effects [25,26]. Structures of GPCRs in complex with biased agonists suggest that distinct receptor conformations are conducive to the binding of select transducers, which in turn mediate distinct signaling events in cells [27]. However, the structural requirements for arrestins to bind their partners and activate individual signaling pathways remain largely unknown. No structure of any arrestin in complex with a non-receptor signaling protein is available today. Here we sought to compare the structural requirements of arrestin-3 for binding to GPCRs and for activating JNK3. We took advantage of arrestin-3 mutants with enhanced or reduced ability to bind GPCRs [12,28,29]. We demonstrate that the structural requirements for receptor binding and arrestin-3-dependent facilitation of ASK1-driven JNK3 activation in cells are dramatically different. These data further support the notion that the JNK pathways are activated by non-receptor-bound arrestin-3. While previous studies suggested this, so far there has been no comprehensive study directly addressing this issue. A practical corollary of our findings is that GPCR ligands, regardless of bias, cannot be used to control this branch of arrestin-3-dependent signaling. Alternative approaches must be developed to allow for therapeutic exploitation of the signaling pathways regulated by free arrestins.

## 2. Materials and Methods

### 2.1. Plasmid Constructs

All constructs used for nanoBiT assays were cloned into pcDNA3 (Invitrogen, Carlsbad, CA, USA). NanoLuc was split into Small BiT (SmBiT, 11 amino acids) and Large BiT (LgBiT, 17.6 kDa), as suggested by Promega (Fitchburg, WI, USA). All arrestin-3 mutants were tagged with SmBiT at the N-terminus with an 11-amino acid linker (SGLKSRRALDS). Human β2AR and M2R were tagged with LgBiT at the C-terminus with a 4-amino acid linker (APAG). The arrestin-3 mutants used in JNK3 activation assays were tagged with Venus at the N-terminus, as described [12].

### 2.2. Cell Culture and Transfection

HEK293 arrestin-2/3 knockout cells (a generous gift of Dr. A. Inoue, Tohoku University, Sendai, Japan) [30,31] were grown in DMEM + GlutaMax (Gibco, ThermoFisher, Waltham, MA, USA) with 10% fetal bovine serum (Gibco, ThermoFisher) and 1% penicillin/streptomycin (Gibco, ThermoFisher) at 37 °C with 5% CO_2_. The absence of arrestin-2/3 in this line was confirmed by western blotting with the pan-arrestin rabbit polyclonal antibody F431 [32]. Cells were transfected using TransHi (FormuMax, Sunnyvale, CA, USA) according to the manufacturer’s instructions (3 µL of TransHi/1 µg of DNA).

### 2.3. In-Cell Arrestin-GPCR Interaction Assay

The NanoBiT assay is based on the complementation of large and small bits fused to different proteins that yield functional luciferase when the two proteins interact. This complementation is reversible [33], which allows the observation of in-cell interactions in real time. The cells were used after no more than ten passages. At 24 h post-transfection, cells expressing similar levels of receptors and arrestins were transferred into a 96-well flat-bottom plate and allowed to adhere in regular culture medium for 4 h. Then the cells were serum-starved overnight (16 h) in culture medium without phenol red. At 48 h post-transfection, luciferase substrate nanoGlo (N1120; Promega, Fitchburg, WI, USA) was added according to the manufacturer’s instructions, and the total luminescence was measured for 20 min using a Synergy Neo plate reader (BioTek, New Castle, DE, USA). Then agonists (10 µM carbachol (carbamoylcholine) for M2R or 10 µM isoproterenol for β2AR) were added, and the luminescence was recorded for 40 min. The expression of arrestin-3 and GPCRs was determined by western blot using anti-arrestin F431 [32] and anti-HA (#3724, Cell Signaling Technology, Danvers, MA, USA) antibodies.

### 2.4. JNK3 Activation Assay

HEK293 arrestin-2/3 KO cells were co-transfected with HA-ASK1, HA-JNK3α2 and either the control (Venus) or indicated N-terminally Venus-tagged form of arrestin-3. After 48 h, cells were lysed with lysis buffer containing 25 mM Tris, pH 7.5, 2 mM EDTA, 250 mM NaCl, 10% glycerol, 0.5% NP-40, 20 mM NaF, 1 mM Na_3_VO_4_, 1 mM phenylmethanesulfonylfluoride (PMSF), 2 mM benzamidine, and a phosphatase inhibitor cocktail (P0044, Sigma, St Louis, MO, USA). Whole cell lysates were centrifuged at 12,000× *g* for 10 min at 4 °C to remove nuclei and cell debris, and the supernatant was used for Western blot analysis. JNK3 activation was measured with the pp-JNK antibody (#4668, Cell Signaling Technology) that recognizes doubly phosphorylated (fully activated) JNK3. The expression of HA-ASK1 and HA-JNK3α2 was determined using an anti-HA antibody (#3724, Cell Signaling Technology). The expression level of Venus and Venus-tagged arrestin-3 proteins was determined with an anti-GFP JL-8 antibody (#632381, Takara Bio USA, San Jose, CA, USA). The endogenous β-actin (loading control) was detected with an anti-actin (#MAB1501, Millipore, Saint Charles, MO, USA) antibody.

### 2.5. Subcellular Localization of Arrestin-3

HEK293 arrestin-2/3 KO cells were co-transfected with HA-ASK1, HA-JNK3α2, and either control (Venus) or indicated N-terminally Venus-tagged forms of arrestin-3 (to mimic the conditions of JNK activation). Twenty-four hours post-transfection, the cells were replated onto poly-D-lysine- and fibronectin-coated Mattek glass bottom dishes. The next day, the medium in the dishes was replaced with 2 mL/dish of FluoroBrite, and the cells’ nuclei were stained with NucBlue reagent (ThermoFisher) according to the manufacturer’s instructions. The cells were then imaged live on the Olympus confocal microscope. Images of between 25 and 45 cells were collected for the analysis with Venus and each form of arrestin-3. The images were analyzed with Nikon NIS-Elements software.

### 2.6. Data Analysis and Statistics

Statistical significance was determined with one-way ANOVA (analysis of variance), followed by Dunnett’s and Bonferroni’s post hoc tests with correction for multiple comparisons using Prism8 software (GraphPad, San Diego, CA, USA). The WT arrestin-3 value groups were used as the comparison group for the Dunnett’s test, unless indicated otherwise. If additional comparisons were of interest, the Bonferroni post hoc test with correction for multiple comparisons of all means was performed, as indicated in the figure legends. In no case perceived outliers were excluded. *p* values < 0.05 were considered statistically significant.

## 3. Results

### 3.1. Functional Role of Arrestin-3 Conformational Equilibrium

To compare global structural requirements for GPCR binding and facilitation of JNK3 activation, we used mutants with significantly perturbed conformational equilibrium (Figure 1). A mutant with a seven-residue deletion in the inter-domain hinge (∆7) was shown to be receptor binding-deficient [12,22]. That was likely because the shortened hinge precludes the twist of the two domains relative to each other invariably detected upon GPCR binding [34], effectively “freezing” the ∆7 mutant in the basal (often referred to as inactive) conformation. We also tested four classes of “enhanced” arrestin mutants with increased receptor binding [28,35,36,37,38]. One was generated by the triple alanine substitution in the C-tail (I386A, V387A, and F388A; 3A mutation), which precluded the anchoring of the arrestin C-tail to the N-domain, thereby promoting receptor binding [35,39,40] (Figure 1A). The deletion of the C-tail (residues 394–409), which yielded truncated arrestin-3-(1–393) (Tr393), had the same effect. The R170E (RE) and D291R (DR) charge reversal mutations destabilize the polar core, another “clasp” holding arrestins in the basal conformation (Figure 1A). The disruption of the polar core also facilitates receptor binding [38,41,42].

To exclude receptor bias, we used two functionally and structurally distinct GPCRs: the Gs-coupled β2-adrenergic receptor (β2AR) and the Gi-coupled M2 muscarinic acetylcholine receptor (M2R). M2R has a very large third intracellular loop that contains all sites phosphorylated in response to receptor activation [43] that are necessary for arrestin binding [44,45]. In contrast, β2AR has a much shorter third loop, and its phosphorylation sites necessary for arrestin binding are localized in the C-terminus [46]. We measured in-cell arrestin-3 binding to these receptors using the nanoluciferase complementation assay (NanoBiT), which was shown to be reversible [33], allowing us to follow in-cell interactions in real time.

Mutants with changed global conformational equilibrium expressed at the same level (Appendix A) showed dramatically different abilities to bind β2AR (Figure 2A). 3A, Tr393, RE, and DR demonstrated greater agonist-dependent recruitment to activated β2AR than WT. In contrast, ∆7 mutant showed no detectable agonist-induced β2AR binding (Figure 2A). Similarly, arrestin-3 mutants with weakened conformational constraints expressed at the same level (Appendix A) demonstrated higher M2R binding than WT, whereas ∆7 mutant did not bind (Figure 2B). These effects cannot be explained by the different levels of mutants’ interaction with unstimulated receptors (Appendix A). Next, we tested the role of conformational equilibrium in arrestin-3-dependent JNK3 activation (Figure 2C). As so far only one MAP3K, ASK1, has been shown to serve as an upstream kinase inducing JNK3 activation [7,47], we used it as an initiator. All arrestin-3 mutants with enhanced GPCR binding showed a much lower ability to facilitate ASK1-induced JNK3 phosphorylation than WT (Figure 2C). The two mutants with detached and deleted C-tail (3A and Tr393, respectively) yielded the lowest JNK3 phosphorylation (Figure 2C). The 3A mutant only marginally increased JNK3 phosphorylation. The ability to facilitate JNK3 activation was completely abolished by the deletion of the C-tail in Tr393 (Figure 2C). Both arrestin-3 mutants with a destabilized polar core (DR and RE) enhanced JNK3 phosphorylation less effectively than WT. In contrast, the ∆7 mutant with severely impaired GPCR binding effectively facilitated JNK3 phosphorylation (Figure 2C).

Upstream MAP3Ks are predominantly cytoplasmic, while GPCRs responding to extracellular agonists are on the plasma membrane. The cytoplasmic arrestin-3 is both recruited to receptors and scaffolds MAPK activation cascades. Thus, if the cytosolic availability of arrestin-3 is altered by the mutations, this might affect the functional properties of the mutants. Arrestin-3 possesses a nuclear export signal localized at its C-terminus [23,48]. WT arrestin-3 is predominantly cytosolic with a low presence in the nucleus (Figure 2D,E). We tested the subcellular localization of arrestin-3 mutants (Figure 2D) and quantified (Appendix A) their distribution in comparison with WT (Venus evenly distributed throughout all cell compartments served as a control) (Figure 2E). The Tr393 mutant, with the arrestin-3 nuclear export signal deleted, was the only one enriched in the nucleus (Figure 2D,E), as compared to WT. Nevertheless, it was still present in the cytoplasm in sufficient quantities to interact with both GPCRs, and was recruited much better than WT (Figure 2A,B). The subcellular localization of 3A, DR, RE, and ∆7 mutants was essentially the same as that of WT (Figure 2D,E). The data show that mutants’ effectiveness in promoting JNK3 activation (Figure 2C) or binding to the receptors (Figure 2A,B) does not correlate with their subcellular localization, suggesting that the conformation of the mutants, rather than their distribution in the cell, determines their effectiveness in both functions. The data clearly show that the ability of different forms of arrestin-3 to bind GPCRs does not correlate with their ability to facilitate JNK3 activation. If anything, these functions appear to have opposite structural requirements.

Structural studies do not provide any clues regarding possible functions of the arrestin C-tail, as the distal C-tail of all arrestin subtypes was not resolved in the available structures [3,41,49,50,51,52,53,54]. The Tr393 arrestin-3 mutant was produced by the deletion of the whole distal C-tail of arrestin-3 (residues 394–409). The effects of this relatively large 15-residue deletion on receptor binding and JNK3 activation were profound and opposite: it greatly increased GPCR binding (Figure 2A,B) but abolished the ability of arrestin-3 to facilitate JNK3 activation (Figure 2C). Therefore, we tested the effects of a series of smaller C-tail deletions (Figure 3A) on both functions. The JNK3 activation was virtually unaffected by the deletion of up to six C-terminal residues but sharply declined upon further truncation (Figure 3B). Removal of up to nine residues from the C-tail did not affect recruitment to M_2_R or β_2_AR (Figure 3C,D). Further deletions increased the binding of mutants expressed at the same level (Appendix A) to both receptors (Figure 3C,D). These data suggest that C-tail residues 401–403 might play an important role in JNK3 activation by full-length arrestin-3. The opposite effects of most C-tail deletions on GPCR binding and JNK activation further demonstrate the difference in structural requirements for these two functions of arrestin-3.

Since the nuclear export signal (NES) is localized to the arrestin-3 C-terminus (Figure 3A), progressive truncations are likely to alter the subcellular distribution of the truncation mutants. We tested the subcellular localization of the series of truncation mutants (Figure 3E,F) and found that progressive shortening of the C-terminus did indeed result in a gradual reduction of their preferential localization to the cytoplasm (Figure 3E). This was particularly evident by the increased nuclear density per pixel (Figure 3F, upper panel). However, the overall cytosolic versus nuclear distribution was significantly altered only in the shortest truncated mutants, Tr389 and Tr393 (Figure 3F). Furthermore, the cytoplasmic concentration of even the shortest mutants was sufficient for enhanced recruitment to the GPCRs localized in the plasma membrane (Figure 3C,D). Also, Tr389, Tr393, and Tr400 demonstrate virtually the same reduction in JNK3 activation (Figure 3B), even though the distribution of Tr400 is more WT-like than that of the two shorter mutants (Figure 3E,F). Thus, the subcellular distribution of truncated mutants does not correlate with their effectiveness in scaffolding the JNK3 activation cascade, suggesting that their conformation plays a more important role, as in the case of structurally distinct mutants shown in Figure 2.

### 3.2. Functional Role of Lariat Loop Lysine

Receptor-attached phosphates were shown to be critical for arrestin binding to most GPCRs both experimentally [28,37,55,56,57] and by modeling [58]. Conserved lariat loop lysine (Lys-294 and Lys-295 in arrestin-2 and -3, respectively) interacts with a phosphate in bound phosphopeptide [42], bound GPCRs [34,59,60,61,62] (Figure 1B), or the putative non-receptor activator abundant cytoplasmic metabolite inositol hexakisphosphate [63]. To test the role of this lariat loop lysine in receptor binding and JNK3 activation, we replaced it with Ala (charge neutralization) or Glu (charge reversal) on the WT background. The positions of all mutations on the linear arrestin-3 sequence are shown in Figure 1C. We found that both K295E and K295A mutants were recruited to β2AR essentially like WT arrestin-3 (Figure 4A). In contrast to β2AR, K295A or K295E slightly increased WT arrestin-3 recruitment to M2R (Figure 5A). Although the rise is much smaller (from ~8 fold to ~13 fold) than that induced by pre-activating mutations (Figure 2B), this difference indicates that the residue at the 295 position plays a role in arrestin-3 recruitment to M2R but not to β2AR (Figure 2). Neither K295A nor K295E mutations appreciably affected the ability of WT arrestin-3 to facilitate JNK3 phosphorylation (Figure 6).

Enhanced arrestin variants with a conformational equilibrium shifted towards a receptor-bound-like state demonstrate increased binding to all functional forms of GPCRs [28,35,37,38]. WT arrestin has to “jump” over a high energy barrier to achieve receptor-binding conformation [64]. Based on experimental data [65] and modeling [58], these mutants have a significantly lower energy barrier. The data obtained with arrestin-1 suggest that the mode of binding of enhanced arrestin mutants to active phosphorylated receptors might be different than that of WT [66]. As mutations that change global arrestin conformation severely affect their ability to activate JNK3 (Figure 2 and Figure 4), the effects of substitution of other functionally important residues on enhanced backgrounds might differ from their effects in WT proteins. Therefore, we tested the effect of K295 mutations on enhanced arrestin-3 backgrounds on binding to both GPCRs and JNK3 activation.

Lys-295 substitutions on the 3A background affected the recruitment to β2AR. K295E slightly decreased the 3A effect (Figure 4B), although the binding of this combination mutant was still much higher than that of WT. K295A completely eliminated the 3A effect (Figure 4B), reducing the binding to the WT level. Similarly, to their effects on β2AR binding, K295A dramatically reduced the recruitment of the 3A mutant to M2R, almost to WT level, while K295E had no effect (Figure 5B). In contrast, on the Tr393 background, both K295A and K295E increased the binding to β2AR to a similar extent (Figure 4C). The effects of K295A and K295E on Tr393 recruitment to M2R were similar, with both mutations further enhancing binding (Figure 5C). As 3A and Tr393 share the same mechanism of arrestin-3 pre-activation, differential effects of K295A and K295E mutations on these two backgrounds suggest that Lys-295 may play a role in the displacement of the arrestin C-tail and/or its repositioning upon receptor binding [67].

Lys-295 is localized on the lariat loop, which supplies two negative charges to the polar core (Asp-291 and Asp-298 in arrestin-3) [3] (Figure 1A). Therefore, we expected Lys-295 substitutions to affect arrestin-3 binding to GPCRs on RE and DR backgrounds, where the polar core is destabilized. However, neither Lys-295 substitution greatly affected the binding of the RE mutant to either GPCR (Figure 4D and Figure 5D). K295A did not significantly change arrestin-3 binding to β2AR on the RE background, whereas K295E slightly reduced it, although the binding was still significantly higher than that of WT (Figure 4D). In the case of M2R, the effects were opposite: K295E did not change the RE binding, whereas K295A slightly reduced it (Figure 5D). On the DR background, K295A had no effect on binding to β2AR, whereas K295E increased arrestin-3-DR recruitment to β2AR (Figure 4E). Both K295A and K295E substitutions greatly increased DR binding to M2R (Figure 5E). These data are consistent with our previous finding that lysine in the lariat loop does not appear to participate in binding receptor-attached phosphates in arrestin-1, -2, or 3 [68]. Yet it contacts one of the phosphates on different GPCRs in most solved structures of receptor-bound arrestin-1 [34,69] and -2 [59,60,61,62,70,71]. Two reasons for this discrepancy are conceivable: one, reengineered rather than wild-type GPCRs were used for structure determination in all cases; two, in many structures, native phosphorylatable elements of GPCRs were replaced with a multi-phosphorylated C-terminal peptide of the vasopressin V2 receptor. Structures of wild-type receptors with phosphates in the positions phosphorylated in vivo with bound cognate arrestins are needed to shed light on this issue.

The two receptors tested are very different structurally (the sizes of their cytoplasmic elements and the position of phosphorylation sites) and functionally (coupling to different G proteins). While the initial binding of arrestin-3 and all mutants to β2AR was followed by rapid dissociation, the complex of M2R with all variants of arrestin-3 appears much more stable than the complex with β2AR (compare Figure 4 and Figure 5). It is tempting to speculate that this difference reflects the absence of a complete “phosphorylation code” in β2AR noted earlier [34]. A high signal even after 60 min of stimulation of M2R rules out the depletion of luciferase substrate during the assay as the reason for the signal decline in the case of β2AR (Figure 2, Figure 3, Figure 4 and Figure 5). One possible explanation is that the complex of arrestin-3 with β2AR is transient [72], so that arrestin-3 dissociates from β2AR upon internalization. It should also be noted that all in-cell assays of the arrestin-receptor interactions (FRET, BRET, and NanoBiT used here) involve the addition of large tags to the interacting proteins. This might be the reason for the different kinetics of the same arrestin-receptor pair observed using different methods. As the NanoBiT assay requires the smallest tags, it is likely that the behavior of the complex with these modifications of interacting proteins is closer to the behavior of the biologically relevant complex of wild-type proteins, although even in this case it is unlikely to be identical to it. In summary, K295 mutations do affect receptor binding. Remarkable similarity in the effects of K295A and K295E mutations on all backgrounds on the binding to these two receptors suggests that Lys-295 plays a role in receptor binding-induced conformational rearrangements of arrestin-3 rather than directly via its interactions with the two structurally different receptors tested.

In contrast to GPCR binding, K295A and K295E substitutions on all backgrounds had virtually no effect on the ability of the mutants to facilitate JNK3 phosphorylation (Figure 6A,B). The only exception was K295E on the 3A background; it partially rescued JNK activation by the 3A mutant, which was ineffective in this regard (Figure 6). This finding is consistent with the idea suggested by the data with truncation mutants (Figure 3) that in the context of full-length proteins, the C-tail plays a role in JNK3 activation. As K295A did not have this effect (Figure 6), it is tempting to speculate that glutamic acid in the 295 position creates an anchor for the C-tail released by the 3A mutation, bringing it into a more favorable position for JNK3 activation. The data show that Lys-295 plays a much greater role in arrestin-3 binding to GPCRs (likely interacting with non-phosphorylated receptor elements [68]) than in its ability to facilitate JNK3 activation, yet again revealing the difference in structural requirements for these two functions.

## 4. Discussion

Non-visual arrestins interact with hundreds of GPCR subtypes and >100 non-receptor signaling proteins [73]. Arrestins selectively bind the active phosphorylated form of their cognate receptors and shut down G protein-mediated GPCR signaling [1], as well as activate numerous signaling pathways (reviewed in [5,74]). Importantly, many proteins interact with free (non-receptor-bound) arrestins [20,21,22,23,24], although the signaling consequences of most of these interactions remain unexplored. Information on the role of the global arrestin conformation and individual structural elements in different arrestin functions is limited. It would help to understand the mechanistic underpinnings of arrestin-dependent signaling and reveal whether arrestin interactions with GPCRs play a role in individual branches of that signaling.

We tested the role of arrestin-3 conformational equilibrium in two functions: GPCR binding and JNK3 activation. Arrestins are held in their basal conformation by two “clasps” [28,37], the polar core composed of several interacting charged residues, and a three-element interaction of β-strand I, α-helix, and β-strand XX of the C-tail mediated by bulky hydrophobic side chains in all three elements [51] (Figure 1A). Upon binding to a GPCR, both of these clasps are broken. A domain twist is observed in all structures of receptor-bound arrestins [34,42,59,60,61,62,63,69,70,71], suggesting that this twist is required for GPCR binding. In the ∆7 mutant, the conformational equilibrium was perturbed by a seven-residue deletion in the 12-residue inter-domain hinge, likely preventing this domain twist and, consequently, precluding receptor binding (Figure 2, Figure 4, and Figure 5). We have previously documented the inability of the ∆7 mutant to bind several GPCRs [22,29,75]. The mutants with enhanced GPCR binding were created by destabilizing the basal conformation. In two mutants, the three-element interaction was disrupted. In 3A, the C-tail is detached by alanine substitutions of bulky hydrophobic residues that keep it anchored to the body of the N-domain; in Tr393, the C-tail is deleted [28,37,76]. We also used polar core mutants R170E and D291R, where the charges of the two residues forming the critical salt bridge in the polar core [3,50,51] were individually reversed. We also tested the role of positively charged Lys-295 implicated in binding to the phosphates attached to the GPCRs (Figure 1B) [34,59,60,61,62] and the putative non-receptor activator of arrestin-3, inositol hexakisphosphate [63]. Lys-295 was replaced with neutral alanine or negatively charged glutamic acid on WT and all mutant backgrounds.

In agreement with previous findings using in vitro direct binding [22] and an in-cell BRET-based interaction assay [12], we found that the ∆7 mutant did not bind GPCRs (Figure 2A,B). In contrast, 3A, Tr, RE, and DR mutants bound both receptors significantly better than WT (Figure 2A,B). WT arrestin-3 and its receptor binding-deficient ∆7 mutant facilitated JNK3 activation in cells overexpressing ASK1 [9,12], whereas mutants with enhanced receptor binding demonstrated reduced ability to facilitate JNK3 activation (Figure 2C). Both 3A and Tr were largely ineffective in this regard (Figure 2C). Polar core mutants RE and DR were active but showed a significantly lower ability to promote JNK3 activation than WT arrestin-3 or ∆7 (Figure 2C). Thus, the effects of mutations changing the conformational equilibrium of arrestin-3 on GPCR binding and JNK3 activation are virtually opposite (Figure 2 and Figure 6). Only the ∆7 mutant effectively “frozen” in the basal conformation activated JNK3 with high efficacy, whereas the mutants with destabilized basal conformation to favor receptor binding were significantly less efficacious. We found that neither receptor binding nor JNK3 activation correlated with the subcellular distribution of the mutants (Figure 2 and Figure 3). The most parsimonious explanation of these observations is that arrestin-3 activates the JNK pathway independently of GPCRs, likely in its basal conformation.

We have previously shown that one of the phosphates of inositol hexakisphosphate engages Lys-295 in a trimer of free arrestin-3, where this metabolite stabilizes the receptor-bound-like conformation of the protomers [63]. Based on this evidence, we suggested that inositol hexakisphosphate acted as a substitute for a GPCR, converting arrestin-3 into an “active-like” conformation, thereby making arrestin-3 a GPCR-independent JNK3 activator [63]. The data presented here contradict this idea. While the substitutions of Lys-295 with alanine and glutamic acid, neither of which would support phosphate binding, on various backgrounds had differential, often profound effects on receptor binding (Figure 4 and Figure 5), these mutations virtually never affected JNK3 activation (Figure 6). The data suggest that receptor-bound-like arrestin-3 conformation is not only unnecessary for efficient JNK activation but also inhibits this function. These results are consistent with our earlier finding that short N-terminal arrestin-3 peptides lacking GPCR-binding elements facilitate JNK3 activation in vitro and in cells [18,19]. JNK3 binds three elements of arrestin-3, localized on both N- and C-domains [16], but only the N-terminal peptides facilitate JNK3 activation [18,19], likely because upstream JNK3-activating kinases also bind in this region [17].

Upon binding to a GPCR, arrestins undergo a significant conformational rearrangement [34,59,60,61,62,69,70,71], and the receptor-bound conformation of arrestins is often called “active” [34,77]. Our data demonstrate that, from a functional standpoint, it is incorrect to consider the basal arrestin conformation inactive. Both conformations are active; arrestins in each conformation perform specific functions. Demonstrated interactions of arrestin-2 and -3 with numerous non-receptor partners [73] suggest that these proteins might regulate many signaling pathways. For most arrestin-binding partners, it remains to be determined whether they prefer the receptor-bound or basal arrestin conformation. There is evidence indicating that the activation of ERK1/2 [8,11,12,14,78], Src [13,14], and focal adhesion kinase [15] by arrestins depends on their binding to GPCRs. The comparison of the GPCR binding and JNK3 activation efficacies of a large set of arrestin-3 mutants suggests that JNK3 activation is facilitated by arrestin-3 in the basal rather than receptor-bound conformation. Thus, the arrestin-3 conformation conducive to scaffolding the JNK3 activation cascade is different from the conformation necessary for receptor binding. The elucidation of the exact arrestin-3 conformation optimal for scaffolding the ASK1-MKK4/7-JNK3 signaling cascade requires co-structures of full-length arrestin-3 with these kinases.

JNK3 activation is not the only GPCR-independent function of arrestins. Both arrestin isoforms bind microtubules; microtubule and GPCR binding are mutually exclusive since the binding sites overlap [22]. Arrestins recruit the E3 ubiquitin ligase Mdm2, which arrestins in the basal conformation bind quite well [23], to the cytoskeleton, facilitating the ubiquitination of microtubule-associated proteins [22]. Thus, free arrestins can change the subcellular localization of signaling proteins and direct their activity towards particular substrates. Such a function has been proposed earlier for arrestin-assisted ERK1/2 activation, which is performed by receptor-bound arrestins [8]. In the basal conformation, the C-tail of all arrestins is anchored to the N-domain via both three-element interaction and the polar core [3,50,51,53,54]. Receptor binding causes the release of the arrestin C-tail, which makes binding sites for clathrin [79,80] and the clathrin adaptor AP2 [80,81] localized on this element more accessible, thereby promoting internalization of the receptor-arrestin complex (reviewed in [82]). However, free arrestin-2 and -3 in the absence of a receptor also bind clathrin [79]. This interaction plays a critical role in the disassembly of focal adhesions independently of receptor activity [21].

The key question regarding any kind of cell signaling is how it is initiated. Arrestins in receptor-bound and free conformations serve as scaffolds for MAPK cascades that activate ERK1/2 and JNK3, respectively. Their function is likely the same as that of all MAPK scaffolds: facilitate signaling initiated by the inputs that activate the upstream-most kinase in the three-tiered cascade.

At this point, the available evidence is insufficient to conclude whether more arrestin functions are GPCR-dependent or GPCR-independent. While the difference between the effects of balanced and G protein- or arrestin-biased GPCR ligands on pathways mediated by active G proteins and receptor-bound arrestins must be profound, neither type of ligand can affect receptor-independent functions of non-visual arrestins. Different tools must be devised to regulate these branches of cellular signaling. In particular, signaling-biased arrestin mutants and monofunctional peptides distilled from multi-functional arrestin proteins should be explored. These mutants and peptides could greatly expand our toolkit for research and, ultimately, for therapy.

## 5. Conclusions

A comprehensive analysis of a large set of arrestin-3 mutants showed that structural requirements for arrestin-3 binding to GPCRs and for arrestin-3-dependent facilitation of the activation of JNK3 are different and, in most cases, virtually opposite. These data strongly suggest that non-receptor-bound arrestin-3 acts as the scaffold of the JNK3-activating cascade, suggesting that GPCR input does not regulate arrestin-3-assisted JNK3 activation in cells.

## Figures and Tables

**Figure 1 cells-12-01563-f001:**
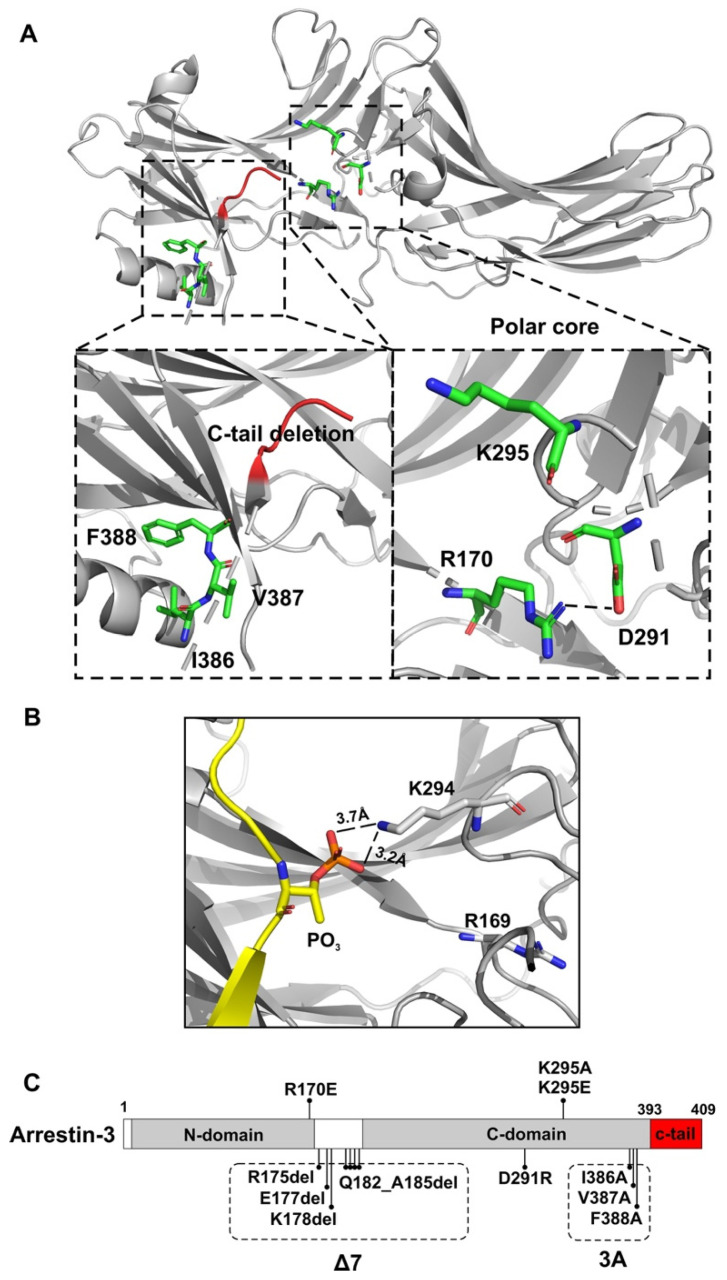
**Arrestin-3 structure and mutations.** (**A**). The crystal structure of arrestin-3 (PDB: 3P2D [3]) with selected elements highlighted to indicate the positions of mutations, which are shown in the insets: **3A:** I386A, V387A, F388A; **RE:** R170E; **DR:** D291R. (**B**). The K294 (corresponding to K295 of arrestin-3) of arrestin-2 interacts with the phosphorylated GPCR C-terminus (shown in yellow; PDB 4JQI [42]. (**C**). Schematic diagram of the arrestin-3 linear sequence with mutations indicated.

**Figure 2 cells-12-01563-f002:**
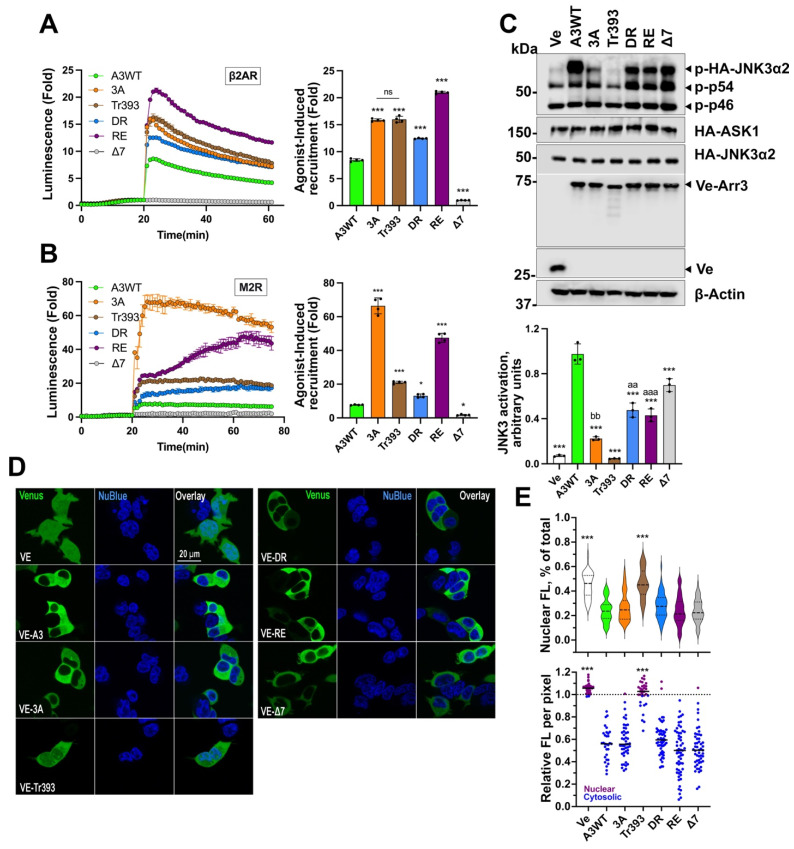
**GPCR binding of arrestin-3 mutants with shifted conformational equilibrium**. The results of the NanoBiT complementation assay (performed as described in Methods) for β2AR (**A**) and M2R (**B**) with WT arrestin-3 and indicated mutants. After the addition of the nanoluciferase substrate, the total luminescence was measured for 20 min until it reached the steady state. Then the agonist (10 μM isoproterenol for β2AR, 10 μM carbachol for M2R) was added, and the luminescence was measured for 40–50 min. Representative results are shown as traces. The receptor binding was normalized to the basal signal without an agonist (read at 20 min). The bars represent the mean ± SEM of four independent experiments performed in triplicate. (**C**) Arrestin-3-dependent JNK3 activation. HEK293 arrestin-2/3 KO cells [30] were co-transfected with Venus or the indicated Venus-tagged arrestin-3 constructs, HA-ASK1 and HA-JNK3α2. Phosphorylation of JNK3 was analyzed by western blot 48 h post-transfection. Upper panel: representative Western blots of phosphor-JNK3α2 and transfected proteins (to show equal expression). Lower panel: Quantification of the phospho-JNK3α2 values. The bars represent means ± SEM of three independent experiments. Data points from each experiment are shown as dots on the bar graphs. The statistical significance of the differences shown, as follows: according to Dunnett’s post hoc comparison to WT arr3: *, *p* < 0.05; ***, *p* < 0.001; or Bonferroni’s post hoc comparison to Δ7: aa, *p* < 0.01; aaa, *p* < 0.001; and to Tr: bb, *p* < 0.01; ns, the difference between the indicated groups is not significant. (**D**) Representative images of the subcellular localization of the arr3 mutants in arrestin-null HEK293 cells. The Venus-tagged arr3 mutants were transfected into cells alongside ASK1 and HA-JNK3 in conditions identical to those used to measure JNK activation (**C**). The cells were counterstained with NucBlue and imaged live on an Olympus confocal microscope. (**E**) The intensity of Venus fluorescence (488 nm) was quantified with Nikon NIS-Elements software and expressed as nuclear fluorescence per pixel (upper panel) or total nuclear fluorescence as a percentage of total cellular fluorescence (lower panel). ***—*p* < 0.001 Dunnett’s post hoc comparison to WT arr3.

**Figure 3 cells-12-01563-f003:**
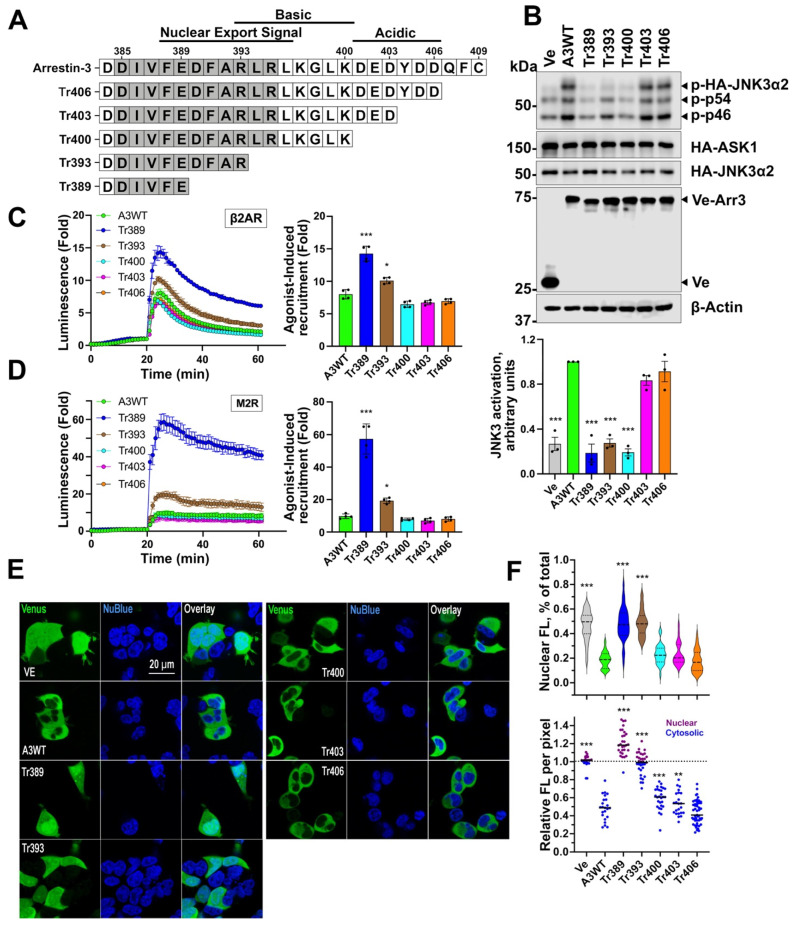
**The effect of C-terminal deletions on arrestin-3-dependent JNK3 activation and GPCR binding**. (**A**) Stepwise deletions in the C-tail of arrestin-3. The nuclear export signal disrupted by ∆393 and ∆389 is indicated. (**B**) Arrestin-3-dependent JNK3 activation is not affected by the deletion of six C-terminal residues but sharply declines thereafter. Upper panel: representative Western blots of phosphor-JNK3α2 and transfected proteins (to show equal expression). Lower panel: Quantification of the phosphor-JNK3α2 values. The bars represent the mean ± SEM of three independent experiments performed in triplicate. Data points from each experiment are shown as dots on the bar graphs. Statistical analysis was performed by one-way ANOVA followed by Dunnett’s post hoc comparison. ***, *p* < 0.001 to WT arr3. Binding of the C-tail deletion mutants with β2AR (**C**) and M2R (**D**) was performed as described in Methods. The bars represent the mean ± SEM of four independent experiments. Data points from each experiment are shown as dots on the bar graphs. The statistical significance of the differences is shown according to Dunnett’s post hoc comparison to WT arr3 as follows: *, *p* < 0.05; ***, *p* < 0.001. (**E**) Representative images of the subcellular localization of the arr3 mutants in arrestin-null HEK293 cells. The Venus-tagged arr3 truncation mutants were transfected into cells alongside ASK1 and HA-JNK3 in conditions identical to those used to measure JNK activation (as in Figure 2C). The cells were counterstained with NucBlue and imaged live on an Olympus confocal microscope. (**F**) The intensity of Venus fluorescence (488 nm) was quantified with Nikon NIS-Elements software and expressed as nuclear fluorescence per pixel (upper panel) or total nuclear fluorescence as a percentage of total cellular fluorescence (lower panel). **, *p* < 0.01; *** *p* < 0.001 Dunnett’s post hoc comparison to WT arr3.

**Figure 4 cells-12-01563-f004:**
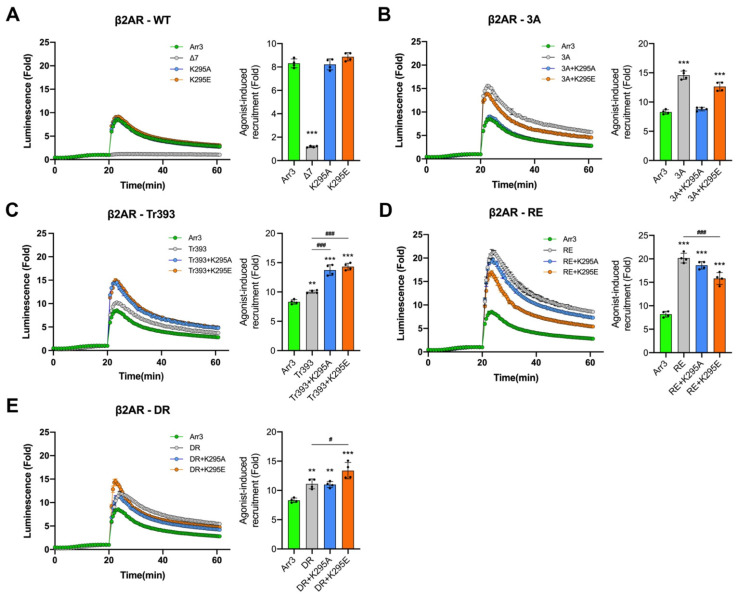
**K295 substitutions yield diverse effects on arrestin-3 recruitment to β2AR.** NanoBiT complementation assay results for β2AR with indicated arrestin-3 mutants: (**A**) WT group; (**B**) 3A group; (**C**) Tr393 group; (**D**) RE group; (**E**) DR group. The curves are colored as follows: arrestin-3 WT, green; pre-activated mutants, gray; +K295A, blue; +K295E, orange. After the addition of the nanoluciferase substrate, the total luminescence was measured for 20 min until it reached the steady state. Then the agonist (10 μM isoproterenol) was added, and the luminescence was measured for 40 min. Representative results are shown as traces. The receptor binding was normalized to the basal signal without an agonist (read at 20 min). The bars represent the mean ± SEM of four independent experiments performed in triplicate. Data points from each experiment are shown as dots on the bar graphs. Statistical significance of the differences is shown as follows: according to Dunnett’s post hoc comparison; **, *p* < 0.01; ***, *p* < 0.001 to WT arr3; or Bonferroni’s post hoc comparison #, *p* < 0.05; ###, *p* < 0.001 between indicated groups.

**Figure 5 cells-12-01563-f005:**
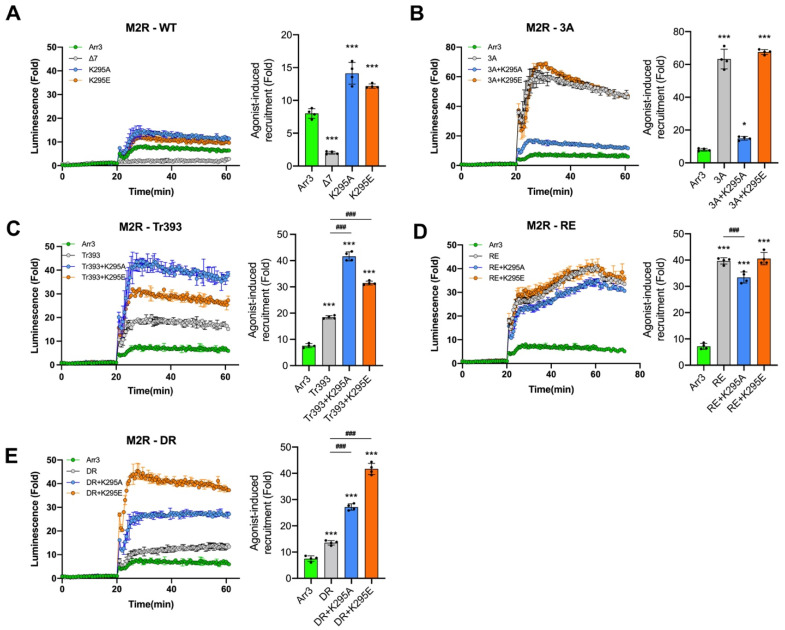
**The effect of K295 substitutions on arrestin-3 recruitment to M2R.** The results of the NanoBiT complementation assay (performed as described in Methods) for M2R with the indicated arrestin-3 mutants: (**A**) WT group; (**B**) 3A group; (**C**) Tr393 group; (**D**) RE group; (**E**) DR group. The curves are colored as follows: arrestin-3 WT, green; pre-activated mutants, gray; +K295A, blue; +K295E, orange. After the addition of the nanoluciferase substrate, the total luminescence was measured for 20 min until it reached the steady state. Then the agonist (10 μM carbachol) was added, and the luminescence was measured for 40 min. Representative results are shown as traces. The receptor binding was normalized to the basal signal without an agonist (read at 20 min). The bars represent the mean ± SEM of four independent experiments performed in triplicate. Data points from each experiment are shown as dots on the bar graphs. The statistical significance of the differences is shown as follows: according to Dunnett’s post hoc comparison *, *p* < 0.05; ***, *p* < 0.001 to WT arr3; or Bonferroni’s post hoc comparison ###, *p* < 0.001 between indicated groups.

**Figure 6 cells-12-01563-f006:**
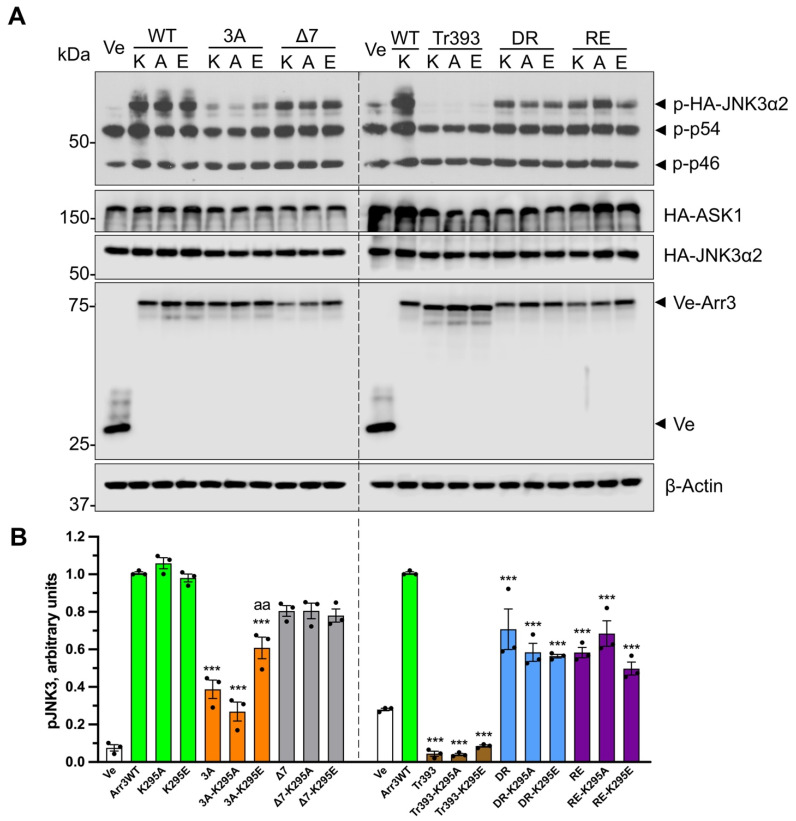
**The effect of K295 substitutions on arrestin-3-dependent JNK3 activation**. (**A**) Western blot analysis of JNK3 activation. HEK293 arrestin-2/3 KO cells [30] were co-transfected with Venus or the indicated Venus-tagged arrestin-3 constructs, HA-JNK3α2 and HA-ASK1. K, A, and E refer to K295, K295A, and K295E. Ve indicates Venus only (no arrestin-3 control). (**B**) Quantification of blots. The relative JNK3 activation was normalized to total JNK3 expression. The bars represent the mean ± SEM of three independent experiments. Data points from each experiment are shown as dots on the bar graphs. The statistical significance of the differences is shown as follows: according to Dunnett’s post hoc comparison ***, *p* < 0.001 to WT arr3; according to Bonferroni’s post hoc comparison aa, *p* < 0.01 to 3A. Note that except for the 3A group, no differences between the parental protein and its K295A and K295E derivates were detected.

## Data Availability

The data are presented in the manuscript. Raw data are available from C.Z. and V.V.G. upon request.

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
