# Peer review of "GPCR Binding and JNK3 Activation by Arrestin-3 Have Different Structural Requirements"

_cells, 2023, doi:10.3390/cells12121563_

Round 1
Reviewer 1 Report
In this manuscript, the authors demonstrate convincingly that JNK3 activation by arrestin3 is reduced when arrestin3 engages a G-protein-coupled receptor. The authors previously published several papers addressing the role of arrestin conformation for JNK3 activation. In this manuscript they add more evidence to show the attenuating role of GPCR engagement by arrestin3 for JNK3 activation, using NanoBiT complementation to assess receptor-arrestin interaction and Western blots for JNK3 activation.
In the NanoBiT complementation assays, the authors often observe that the signal reaches a maximum 2-3 minutes after the beginning of stimulation and thereafter declines, sometimes considerably. This behavior is not seen in other assays of GPCR-arrestin interaction such as BRET or FRET assays. Assuming that the decline is not a result of substrate depletion, it is hard for me to reconcile the differences. Could it be that the amount of receptor-arrestin complex decreases as the receptor is internalized? If this was the case, what would happen if internalization was inhibited?
Some of the data presented by the authors was quite surprising for me, especially the role (or lack thereof) of Lys295 for GPCR binding. The analogous Lys294 in arrestin2 is engaged in phosphopeptide binding in several (but not all, e.g. 7DFA) published crystal structures of phosphorylated GPCRs. However, it appears that the basicity of this residue is dispensable for GPCR interaction under wild-type conditions, and mutation to Ala or Glu often improves arrestin binding to the receptor (rather than attenuating it) in various arrestin mutants. Unfortunately, the authors decided not to address this unexpected finding in their discussion.
Fig. 4C: there seems to be a mislabeling of the bar graph data as “Tr393+K295E” appears twice.
Author Response
In this manuscript, the authors demonstrate convincingly that JNK3 activation by arrestin3 is reduced when arrestin3 engages a G-protein-coupled receptor. The authors previously published several papers addressing the role of arrestin conformation for JNK3 activation. In this manuscript they add more evidence to show the attenuating role of GPCR engagement by arrestin3 for JNK3 activation, using NanoBiT complementation to assess receptor-arrestin interaction and Western blots for JNK3 activation.
We are grateful to the reviewer for these encouraging remarks.
In the NanoBiT complementation assays, the authors often observe that the signal reaches a maximum 2-3 minutes after the beginning of stimulation and thereafter declines, sometimes considerably. This behavior is not seen in other assays of GPCR-arrestin interaction such as BRET or FRET assays. Assuming that the decline is not a result of substrate depletion, it is hard for me to reconcile the differences. Could it be that the amount of receptor-arrestin complex decreases as the receptor is internalized? If this was the case, what would happen if internalization was inhibited?
The reviewer is right to notice that different arrestin-GPCR interaction assays yield different kinetics and to provide one plausible explanation for the kinetics we observed. As dissociation upon internalization should affect the readout of all assays in the same way, we do not believe that inhibition of endocytosis would clarify the issue of assays differences. One possible reason might be tagging receptors and arrestins with large reporters. The tags used in NanoBiT assay are smaller than those used in FRET- or BRET-based assays, but even in this case interacting proteins are not exactly wild type. We expanded the discussion of these points.
Some of the data presented by the authors was quite surprising for me, especially the role (or lack thereof) of Lys295 for GPCR binding. The analogous Lys294 in arrestin2 is engaged in phosphopeptide binding in several (but not all, e.g. 7DFA) published crystal structures of phosphorylated GPCRs. However, it appears that the basicity of this residue is dispensable for GPCR interaction under wild-type conditions, and mutation to Ala or Glu often improves arrestin binding to the receptor (rather than attenuating it) in various arrestin mutants. Unfortunately, the authors decided not to address this unexpected finding in their discussion.
Thanks for pointing out this omission! One possible source of discrepancy is the use of drastically reengineered receptors, often with native phosphorylatable elements replaced with hyper-phosphorylated C-terminal peptide of vasopressin V2 receptor, for structure determination. We expanded the discussion of this important issue.
Fig. 4C: there seems to be a mislabeling of the bar graph data as “Tr393+K295E” appears twice.
Thanks for pointing out this error! The labeling in Fig. 4C was corrected.
Reviewer 2 Report
The present study addressed the structural requirements of arrestin 3 for its interaction/activation with agonist-stimulated GPGRs and the GPCR-independent scaffolding of JNK3, which is functionally involved in its activation by upstream kinases. It comes after numerous previous articles investigating these issues, many of which are from the same authors.
The experiments are well explained, well conducted and contain the appropriate controls.
The study concludes that GPCR binding and arrestin-3-assisted JNK3 activation have distinct structural requirements, consistent with a model where JNK3 activation involves a GPCR-unbound arrestin-3. This concept is not novel and has been addressed (more or less in deep) for many other interacting partners. In some cases the modulatory effect of arrestin 3 is clearly disconnected from the activation of a cognate GPCR.
There are general limits of all studies using an approach based on arrestin mutants characterized for specific gain / loss of function, relative to the functional interaction with a GPCR and a given interacting partner. The need for exogenously expressed constructs, which perturb the equilibrium between the studied scaffolding protein (arrestin 3 in this case) and the many other endogenous partners, probably affect the readout of the experiments without any possible control. For example the mutants with absent NES even if the “leave behind” in the cytoplasm sufficient arrestin 3, they will also displace into the nucleus part of JNK3 and possibly part of JNK3 activation machinery.
I agree with the authors’ statement that a more comprehensive understanding of arrestin conformations, which are optimal for the scaffolding of a given signaling cascade, requires co-structures.
Author Response
The present study addressed the structural requirements of arrestin 3 for its interaction/activation with agonist-stimulated GPGRs and the GPCR-independent scaffolding of JNK3, which is functionally involved in its activation by upstream kinases. It comes after numerous previous articles investigating these issues, many of which are from the same authors.
The experiments are well explained, well conducted and contain the appropriate controls.
We are grateful to the reviewer for encouraging remarks.
The study concludes that GPCR binding and arrestin-3-assisted JNK3 activation have distinct structural requirements, consistent with a model where JNK3 activation involves a GPCR-unbound arrestin-3. This concept is not novel and has been addressed (more or less in deep) for many other interacting partners. In some cases the modulatory effect of arrestin 3 is clearly disconnected from the activation of a cognate GPCR.
The reviewer is correct, this is the main point of our comprehensive study. Previously published data suggested that JNK3 activation is facilitated by non-receptor-bound arrestin-3. The novelty of this study is direct demonstration that structural requirements for this function and GPCR binding are different, virtually opposite.
There are general limits of all studies using an approach based on arrestin mutants characterized for specific gain / loss of function, relative to the functional interaction with a GPCR and a given interacting partner. The need for exogenously expressed constructs, which perturb the equilibrium between the studied scaffolding protein (arrestin 3 in this case) and the many other endogenous partners, probably affect the readout of the experiments without any possible control. For example the mutants with absent NES even if the “leave behind” in the cytoplasm sufficient arrestin 3, they will also displace into the nucleus part of JNK3 and possibly part of JNK3 activation machinery.
Thanks! Indeed, these issues are unlikely to be resolved by a single study, however comprehensive. Unfortunately, one needs to express “reporting” versions of proteins of interest for any in-cell measurements (as opposed to “silent” wild type ones endogenously expressed by the cell), which inevitably results in changes of natural stoichiometry, etc.
I agree with the authors’ statement that a more comprehensive understanding of arrestin conformations, which are optimal for the scaffolding of a given signaling cascade, requires co-structures.
Yes, co-structures of arrestins with signaling proteins other than receptors are needed. Unfortunately, none are available so far.